# Kcr-FLAT: A Chinese-Named Entity Recognition Model with Enhanced Semantic Information

**DOI:** 10.3390/s23041771

**Published:** 2023-02-04

**Authors:** Zhenrong Deng, Yong Tao, Rushi Lan, Rui Yang, Xueyong Wang

**Affiliations:** 1Guangxi Key Laboratory of Images and Graphics Intelligent Processing, Guilin University of Electronic Technology, Guilin 541004, China; 2Nanning Research Institute, Guilin University of Electronic Technology, Guilin 541004, China; 3Guilin Xintong Technology Co., Ltd., Guilin 541004, China

**Keywords:** Chinese-named entity recognition, semantic information, syntactic information, cross-transformer, regularity perception

## Abstract

The performance of Chinese-named entity recognition (NER) has improved via word enhancement or new frameworks that incorporate various types of external data. However, for Chinese NER, syntactic composition (in sentence level) and inner regularity (in character-level) have rarely been studied. Chinese characters are highly sensitive to sentential syntactic data. The same Chinese character sequence can be decomposed into different combinations of words according to how they are used and placed in the context. In addition, the same type of entities usually have the same naming rules due to the specificity of the Chinese language structure. This paper presents a Kcr-FLAT to improve the performance of Chinese NER with enhanced semantic information. Specifically, we first extract different types of syntactic data, functionalize the syntactic information by a key-value memory network (KVMN), and fuse them by attention mechanism. Then the syntactic information and lexical information are integrated by a cross-transformer. Finally, we use an inner regularity perception module to capture the internal regularity of each entity for better entity type prediction. The experimental results show that with F1 scores as the evaluation index, the proposed model obtains 96.51%, 96.81%, and 70.12% accuracy rates on MSRA, resume, and Weibo datasets, respectively.

## 1. Introduction

With the popularity of smart mobile terminals, the number of internet users in developing countries is growing rapidly, and on 31 August, China’s internet Network Information Center (CNNIC) released its 50th statistical report on the development status of China’s internet. According to the report, as of June 2022, there were 1.051 billion internet users in China; a large number of China’s text data will be generated when people obtain information, search for content, and interact with information on the internet. How to make full use of large amounts of Chinese text (to quickly extract and mine valuable information from it) is a popular research direction.

To improve the processing speed of Chinese text, many applications in the field of Chinese natural language processing (NLP) need the support of Chinese NER, including intelligent recommendations [1], question-answering [2], text understanding [3,4], and text generation [5]. Chinese NER is one of the most important and basic tasks in NLP, and it plays an important role in many downstream tasks. Chinese NER mainly identifies entities from raw text and classifies the detected entities into a pre-defined category, such as people names, place names, organizations, and other proper nouns [6]. For example, in Chinese, “*Dong ao hui zai bei jing shun li ju xing.* (The Winter Olympic Games were successfully held in Beijing)”, in which the named entities include “*Dong ao hui* (Winter Olympic Games)” and “*bei jing* (Beijing)”.

Compared to English NER, Chinese NER is more difficult for several reasons. First, the development of Chinese NER is relatively late compared to the development of English NER, the corpus of Chinese NER is less, which restricts the development of Chinese NER [7,8]. Second, there are spaces between English words as separators, so that preprocessing word segmentation is not necessary. However, Chinese does not have these obvious features [9], The formation of the Chinese word is flexible. A Chinese sequence can be divided into different words according to different contexts. For example, in “*huang he nan an* (south bank of the Yellow River)”, the entity is “*hunag he* (Yellow River)”, but in other contexts, “*he nan* (Henan province)” is a place name as another named entity.

To address the aforementioned issues and handle the complexity of the Chinese language structure, this paper presents a Kcr-FLAT for Chinese NER. We use the Flat-Lattice transformer (FLAT) model as the base model to take advantage of its lightweight and parallel computing. Based on word enhancement, the proposed model extracts three different types of syntactic data and corresponding context features, encodes the syntactic information with their context features by a KVMN, and fuses them by attention mechanism. To solve the segmentation error introduced by lexical information, we used a cross-transformer to fully integrate lexical information and syntactic information. Finally, we explored the internal regularity of each entity to improve the recognition accuracy of the named entities through an inner regularity perception module. The main contributions of the proposed model include:We propose a Chinese NER model Kcr-FLAT, which can achieve better performance in three Chinese datasets—MSRA, resume, and Weibo.We extract and encode three types of syntactic data and fuse them with lexical information to solve the segmentation error.We combine the internal regularity of the entity mentions by an inner regularity perception module with external information to improve the performance of the designed Kcr-FLAT model.

The remainder of this article is organized as follows. Section 2 describes the related work of our model. Section 3 introduces the specific composition structure of Kcr-FLAT. Section 4 describes the datasets, and evaluates the index and experimental settings. Section 5 presents the experimental results in detail. Finally, the article is summarized in Section 6.

## 2. Related Works

Ever snce NER was proposed as a subtask at the sixth Message Understanding Conference (MUC-6) in 1995, this topic has attracted the attention of many experts and scholars in China and abroad, and has become a clear research task in the field of NLP. There are three main development stages of NER: rule-based NER, statistical machine learning-based NER, and deep learning-based NER.

The early and mid-term mainly involve rule-based NER and statistical machine learning-based NER. The rule-based method mainly involves naming rules manually, and then matching the characters inside the rules by custom rules to find the entities inside the text [10]; it relies heavily on manual efforts and takes a lot of time to build up the relevant knowledge base and lexicon. It is difficult to match the entities in all scenarios and it has very large scenario limitations without migration. Among the NER techniques in Chinese, Wang et al. [11] formulated named entity rules related to company names for datasets in the financial news domain, considered the structural features of company names and the contextual semantic associations in the text files, and established six knowledge bases for recognizing company names in the financial news domain. To improve the efficiency of specifying rules, scholars began to study how to formulate named entity rules through machine assistance and implemented named entity recognition through unsupervised classification of a large number of datasets [12]. Rule-based named entity recognition methods require a lot of human and material resources and specified rules have great limitations. There are often named entity rules in a certain application field, with low generalization ability.

Statistically-based named entity recognition methods are mainly implemented by machine learning algorithms; the most frequently used in named entity recognition are maximum entropy model (ME) [13], hidden Markov model (HMM) [14], support vector machine (SVM) [15], conditional random field (CRF) [16], etc. Statistical-based named entity recognition methods require a large amount of corpus support and manual set-ups of feature templates that match the specific application scenario.

With the development of neural networks, deep learning methods have become research hotspots. For Chinese NER, various lexicon-based models have been proposed that incorporate external lexicon information and obtain better results. A typical method is Lattice-LSTM [17], which incorporates word lexicons into the character-based NER model. However, limited by the construction method, the lexical representation can only be added to the last word and the lattice structure fails to compute in parallel. To address this problem, Sui et al. [18] proposed a collaborative graph network-based lexical knowledge to achieve automatic lexicon construction through large-size automatic segmentation text pre-training and graph convolution. In the same year, Ding et al. [19] proposed a named entity recognition method based on graph neural networks and combining multiple lexicons, allowing the model to automatically learn the features of lexicons, which helps to alleviate the problem of false matches.

These methods are necessary to manually constructed lexicon resources, which are expensive and time-consuming, and the quality of the lexicon is critical to the tasks. Xu et al. [20], inspired by these works, proposed a character-level word embedding Chinese-named entity recognition model, which enables semantic information to be utilized at multiple granularities from the root, character, and word level. Ma et al. [21] improved Chinese word separation accuracy by merging lexical information into a vector representation, thus avoiding the introduction of complex sequence structures to characterize lexical information. Hu et al. [22] proposed the SLK-NER Chinese-named entity recognition model, which still faces the challenge of matching word boundary conflicts due to the simple first-order lexical knowledge that provides insufficient lexical information, and the authors proposed new insights into the second-order lexical knowledge of each character in a sentence to provide more lexical information, including semantic and word boundary features.

Several works focus on glyph structure information in Chinese NER. Meng et al. [23] proposed the Glyce network, which is the first model to apply Chinese glyph information in the field of Chinese-named entity recognition, improving the generalization ability of computer vision to text data. Wu et al. [24] extracted the radicals of Chinese characters through convolutional neural networks and then incorporated the radicals and character information into the model through a cross-transformer module and random attention mechanism to improve the accuracy of Chinese-named entity recognition. In these works, the structural components of Chinese characters have been proven to enrich the semantics of the characters, which achieved better results without using any external word embeddings and word resources. However, there are two major drawbacks of the glyph models. First, the glyph information codes independently. The interactive knowledge between the glyph and context is ignored. Second, the entity type and quantity of modern Chinese are far richer and more complex than those of ancient Chinese, it has been proven that historical texts are meaningless to NER, to some extent [25].

Recently, more Chinese characters and text features have been used in NER. Nie et al. [26] improved named entity recognition by using the syntactic information with the attentive ensemble. The RICON [27] model was published by Huawei Cloud in 2022, which enhances the prediction of entities by exploring the naming regularity within the entities. In the above work, the Chinese-named entity recognition has reached a very high accuracy rate, which also represents a more mature period for the development of the Chinese-named entity. Thus, it is more desirable to use these semantic data and effectively integrate them together.

## 3. Proposed Method

The overall structure of the proposed Kcr-FLAT model is shown in Figure 1, which mainly consists of three modules, i.e., the KVMN module for encoding the syntactic information with its context features, the cross-transformer module for incorporating lexical information and syntactic information, and the regularity perception module for analyzing inner regularity.

### 3.1. KNMN for Syntactic Embedding

Considering Chinese NER tasks, the character-level NER methods are usually superior to word-level NER methods [28,29,30,31]. In this paper, the character-level NER method is used. Chinese characters may have different meanings in different contexts, and there is no separation between characters, which makes it difficult to obtain entity boundary information. Since lexical boundaries usually play crucial roles in entity boundaries; this paper uses lexical information as external information to enhance the input to obtain entity boundary information. Although the introduction of lexical information does not require splitting the input, it is still affected by segmentation errors. For example, “*huang he liu jin zhong guo duo ge sheng fen* (The Yellow River flows through many provinces in China)”, where “*huang he liu jin zhong guo* (the Yellow River flows through China)” can be decomposed into “*huang he* (Yellow River)”, “*he liu* (river)”, “*liu jin* (flow through)” and so on. However, after syntactic analysis, due to the syntactic constraint, “*huang he* (Yellow River)” and “*zhong guo* (China)” are the inherent names, and “*liu jin* (flow through)” is the only verb, so it is easy to be correctly classified as “*huang he* (Yellow River) + *liu jin* (flow through)”, eliminating the ambiguity caused by the word “*he liu* (river)”. Therefore, this paper introduces syntactic information to eliminate ambiguity; it encodes different types of contextual features and their syntactic data, including POS labels [32], syntactic constituents [33], and dependency relations [34], to improve the recognition accuracy of named entities in different contexts. To obtain the embedding of syntactic information, the input sequence *X* is firstly parsed using off-the-shelf NLP parsing tools to obtain three types of syntactic data and corresponding text features [26], the extraction of syntactic information is illustrated in Figure 2.

Each syntactic information is encoded using KVMN and integrated by the attention mechanism to obtain the feature embedding of Chinese characters at the sentence level. The encoding of syntactic information of “*gui lin dian zi ke ji da xue zuo luo yu guang xi.* (Guilin University of Electronic Science and Technology is located in Guangxi)” is shown in Figure 3.

After parsing the input sequence *X*, each character $x_{i}$ in *X* is contained in a word, using the word as the center word and mapping its context features and syntactic information into a set of keys and values denoted by $K_{i}^{c}=\left[k_{i,1}^{c},\ldots ,k_{i,j}^{c},\ldots ,k_{i,m_{i}}^{c}\right]$ and $V_{i}^{c}=\left[v_{i,1}^{c},\ldots ,v_{i,j}^{c},\ldots ,v_{i,m_{i}}^{c}\right]$, respectively. Where c∈C={P,S,D}, P,S,D denotes a syntactic type respectively, $m_{i}$ is the number of text features of $x_{i}$ in syntactic type *c*, $k_{i,j}^{c}$ is the *j*th context feature of $x_{i}$ for the syntactic type *c*, and $v_{i,j}^{c}$ is the syntactic information of $k_{i,j}^{c}$. Then mapping $K_{i}^{c}$ and $V_{i}^{c}$ into matrix form as $e_{i,j}^{k_{c}}$ and $v_{i,j}^{k_{c}}$. Therefore, $e_{i,j}^{k_{c}}$ refer to the context features of $x_{i}$, $v_{i,j}^{k_{c}}$ refer to the corresponding syntactic data. The syntactic information is calculated by
(1)$$p_{i,j}^{c}=\frac{\exp\left(h_{i}\cdot e_{i,j}^{k_{c}}\right)}{\sum_{j=1}^{m_{i}}\exp\left(h_{i}\cdot e_{i,j}^{k_{c}}\right)}$$
where $p_{i,j}^{c}$ is the weight assigned to the syntactic information. Since Chinese syntax is word-based and multiple characters in the same word share the same syntactic information, For example, “*da xue* (university)”, in which “*da*” and “*xue*” have the same vector encoded by syntactic information; therefore, $h_{i}$ is introduced to distinguish the representation of different characters in the same word, $h_{i}$ is obtained by
(2)$$h_{i}=E\left(x_{i}\right)$$
where *E* is the character-level embedding layer. Then, the weights $p_{i,j}^{c}$ were assigned to the corresponding syntactic data $v_{i,j}^{k_{c}}$, which is computed by
(3)$$s_{i}^{c}=\sum_{j=1}^{m_{i}}p_{i,j}^{c}e_{i,j}^{v_{c}}$$
where $s_{i}^{c}$ is the output of KVMN, containing the weighted syntactic information of type *c*. Therefore, the syntactic data are weighted and encoded according to their corresponding context features so that useful information could be leveraged accordingly.

After the syntactic information vector $s_{i}^{c}$ is obtained, to integrate the three types of syntactic data and resolve the conflicts between different syntactic information, different weights are assigned to each syntactic information. The different syntactic information is integrated through the attention mechanism. The weight $q_{i}^{c}$ of the syntactic information $s_{i}^{c}$ is calculated by
(4)$$q_{i}^{c}=\sigma \left(W_{q}^{c}\cdot \left(h_{i}⊕s_{i}^{c}\right)+b_{q}^{c}\right)$$
where *W* is the trainable vector, *b* is the bias, ⨁ is the concatenate operation, and σ is the *sigmoid* function. The attention of the syntactic information is subsequently calculated using the *softmax* function, and the attention is calculated by
(5)$$a_{i}^{c}=\frac{\exp\left(q_{i}^{c}\right)}{\sum_{c\in C}\exp\left(q_{i}^{c}\right)}$$
where $a_{i}^{c}$ is the attention of the corresponding syntactic information of type *c*. Finally, the attention is assigned to the corresponding syntactic information and the three different types of syntactic data are fused into $s_{i}$ by
(6)$$s_{i}=\sum_{c\in C}a_{i}^{c}s_{i}^{c}$$

Compared with simply connecting the three kinds of syntactic information, the attention mechanism selectively leverages different features, resolving the conflicts between different syntactic data by distinguishing more important ones from others. Therefore, different types of syntactic data are comprehensively and selectively encoded into $s_{i}$.

### 3.2. Cross-Transformer for Semantics Fusion

Since lexical information is more concerned with the relationship between words and characters, it is easier to identify local information, such as word position and boundaries, while syntactic information is more concerned with the overall information of the sentence, and the segmentation errors introduced by lexical information are corrected by syntactic constraints. After obtaining the embedding of syntactic information, the lexical and syntactic data are fused using a cross-transformer. The cross-transformer network is shown in Figure 4.

The input ($Q_{x}^{L}$, $K_{x}^{L}$, $V_{x}^{L}$) of the left transformer encoder is obtained from lattice embedding, which contains lexical information by linear transformation as follows:(7)$$\left[Q_{x}^{L},K_{x}^{L},V_{x}^{L}\right]=E_{x}^{L}\left[W_{Q}^{L},W_{K}^{L},W_{V}^{L}\right]$$
where $E_{x}^{L}$ is the lattice embedding, $E_{x}^{L}\in \left\{x_{1},\ldots ,x_{i},\ldots ,x_{N}\right\}$, $x_{i}$ is the lexical representation of the input and *N* is the length of the input sequence *X*, each $W^{L}$ is a learnable parameter. The $Q_{s}^{R}$, $K_{s}^{R}$, and $V_{s}^{R}$ of the left transformer encoder are obtained from the embedding of syntactic information and by linear transformation as follows:(8)$$\left[Q_{s}^{R},K_{s}^{R},V_{s}^{R}\right]=E_{s}^{R}\left[W_{Q}^{R},W_{K}^{R},W_{V}^{R}\right]$$
where $E_{s}^{R}$ is the syntactic embedding, $E_{s}^{R}\in \left\{s_{1},\ldots ,s_{i},\ldots ,s_{N}\right\}$, $s_{i}$ is the syntactic representation of the input, we keep the length of the input sequence at *N* to ensure that the input of the cross-transformer has the same length at both ends; each $W^{R}$ is a learnable parameter. The cross-transformer consists of two transformer encoders. Each encoder is composed of self-attention and feed-forward network (FFN) layers, followed by residual connection and layer normalization. FFN is a position-wise multi-layer perceptron (MLP) with the nonlinear transformation of the semantic space. The self-attention layer is used to extract semantic-level information. The attention score of the vanilla transformer is calculated as:(9)Att(A,V)=softmax(A)V
(10)$$A_{i,j}=\left(\frac{Q_{i}K_{j}^{T}}{\sqrt{d_{k}}}\right)$$
where $d_{k}$ is the dimension of *K*. To cross the syntactic information and lattice information obtained from Equations (7) and (8), we use a variant of self-attention [35] with the relative position encoding in the FLAT model, which is computed as follows:(11)$${Att}^{L}=softmax\left(A^{R}\right)V^{L}$$
(12)$${Att}^{R}=softmax\left(A^{L}\right)V^{R}$$
where $A^{R}$ is the syntactic attention score and $A^{L}$ is the lattice attention score. $A^{R}$ is computed by
(13)$$A_{i,j}^{R}={\left(Q_{i}^{R}+u^{R}\right)}^{T}K_{j}^{R}+{\left(Q_{i}^{R}+v^{R}\right)}^{T}\left(R_{i,j}^{R}W_{r}^{R}\right)$$
where $u^{R}$ and $v^{R}$ are attention bias, $W_{r}^{R}$ are learnable parameters. $R_{i,j}^{R}$ is the relative position encoding, which is computed by
(14)$$R_{i,j}^{R}=ReLU\left(W_{r}\left(p_{h_{i}-h_{j}}⊕p_{t_{i}-h_{j}}⊕p_{h_{i}-t_{j}}⊕p_{t_{i}-t_{j}}\right)\right)$$

The relative position encoding $R_{i,j}$ is used to avoid the loss of directionality caused by the inner dot-product of the vector. Each *p* denotes a relative distance. The calculation of ${Att}^{L}$ is essentially the same.

### 3.3. Regularity Perception Module

In the patterns of the word formation of Chinese and its language structure, there are category-to-category named patterns for entities, e.g., “XX-textitxue xiao (school)” and “XX-*fa yuan* (court)” usually represent place names, but also regularity within words, e.g., “*ma nao he* (agate river)”, “*huang he* (yellow river)”, “*liu yang he* (liuyang river)”, etc., are all in the format of “XX-*he* (river)” to express locations. To capture the regularity of these entities, this paper uses the regularity perception module proposed by Huawei Cloud in 2022 to analyze the internal regularity. For the NER task, there are two classical solutions, one is to handle it as a sequence-labeling task, but sequence labeling does not easily identify nested entities, e.g., “*gui lin dian zi ke ji da xue* (Guilin University of Electronic Science and Technology)”, which identifies the complete “Guilin University of Electronic Science and Technology” but not a place entity “*gui lin* (Guilin) and a textitdian zi ke ji da xue (University of Electronic Science and Technology of China) which means a totally different university”. This problem can be effectively avoided by using the span method, which can be used to search, reorganize, and classify words on the span. In this paper, we adopt the span method to process the named entity, use the regularity perception module to capture the internal regular features of each span $s_{i,j}$, and replace the linear and CRF at the top of the cross-transformer with the regularity perception module, as shown in Figure 5.

In Figure 5, the regularity representation of each span is obtained by a linear attention layer calculated as follows:(15)$$h_{s_{i,j}}^{rec}=\sum_{t=i}^{j}\alpha_{t}\cdot h_{t}$$
where $h_{t}$ is the concatenated output from cross-transformer, $t\in \left\{i,i+1,...,j\right\}$ refers to the index of span $s_{i,j}$, $\alpha_{t}$ is the attention score of $h_{t}$, which is calculated as follows:(16)$$\alpha_{t}=\frac{exp\left(a_{t}\right)}{\sum_{k=i}^{j}exp\left(a_{k}\right)}$$
(17)$$a_{t}=W_{reg}^{T}h_{t}+b_{reg}$$
where $W_{reg}^{T}\in R^{d\times 1}$ and $b_{reg}\in R^{1}$ denote the learnable weight and bias, respectively. The model uses a biaffine attention mechanism to learn the interactions of head and tail features to integrate the regularity between entities into the representation of span which is computed by
(18)$$h_{s_{i,j}}^{\left(span\right)}=h_{i}^{T}U^{\left(1\right)}h_{j}+\left(h_{i}⊕h_{j}\right)U^{\left(2\right)}+b_{1}$$
where $h_{i}$ and $h_{j}$ are the head and tail representations of the span $s_{i,j}$, respectively, $U^{\left(1\right)}\in R^{2d\times 2d\times 2d}$ and $U^{\left(2\right)}\in R^{4d\times 2d}$, integrating the regularity and span representation through a gate network:(19)$$g_{s_{i,j}}=\sigma \left(U^{\left(3\right)}\left[h_{s_{i,j}}^{\left(span\right)};h_{s_{i,j}}^{\left(reg\right)}\right]+b_{2}\right)$$
(20)$$h_{s_{i,j}}=g_{s_{i,j}}⊙h_{s_{i,j}}^{\left(span\right)}+\left(1-g_{s_{i,j}}\right)⊙h_{s_{i,j}}^{\left(reg\right)}$$
where $U^{\left(3\right)}\in R^{4d\times 1}$ is the learnable weight, $b_{2}$ is the bias, σ is the *sigmoid* function, ⊙ refers to the element-wise multiplication operation. Then, the prediction of the entity type for each span is obtained from a softmax linear classifier:(21)$${\tilde{y}}_{s_{i,j}=Softmax\left(W_{type}^{\left(T\right)}h_{s_{i,j}}+b_{3}\right)}$$
where $W_{type}^{\left(T\right)}\in R^{2d\times c}$ is the trainable weight and $b_{3}$ is the bias. The loss function is represented by the cross entropy:(22)$$L_{aware}=-\frac{1}{n}\sum_{n=1}^{N}\sum_{i=1}^{l}\sum_{j=1}^{l}y_{s_{i,j}}^{\left(n\right)}log\left({\tilde{y}}_{s_{i,j}}^{\left(n\right)}\right),i\leq j$$
where ${\tilde{y}}_{s_{i,j}}^{\left(n\right)}$ is the predicted type of span, $y_{s_{i,j}}^{\left(n\right)}$ is the ground truth type of span, and *N* is the number of training samples in the regularity perception module.

## 4. Experimental Settings

In this section, we evaluate the proposed Kcr-FLAT method on three Chinese datasets. We use the span method to calculate F1-score (F1), precision (P), and recall (R) as the evaluation metrics.

### 4.1. Datasets

In this paper, we use three Chinese NER datasets to evaluate our model, including MSRA [36], resume [17], and Weibo [37], respectively. The MSRA dataset comes from the Third International Chinese Language Processing Bake-Off: word segmentation and named entity recognition, which mainly contains news datasets and is one of the most widely used datasets in the NER task. The resume dataset was proposed by Zhang et al.; it contains a large number of names of people, places, and companies. The Weibo dataset was proposed by Peng et al.; it contains named entities for multiple scenarios from micro-blogging and can extend the application of named entity models. Among them, the Weibo dataset has four types of entities—PER (person), ORG (organization), LOC (location), and GPE (government-wide point of entry). Resume has eight types of entities—CONT (country), EDU (education), LOC, PER, ORG, PRO (profession), RACE (race), and TITLE (job title). The MSRA dataset contains three types of entities—ORG, PER, and LOC.

### 4.2. Evaluation Index

The weighted summed average *F*1 is used as the evaluation index in this paper. *F*1 is calculated as:(23)$$F1=\frac{2\times P\times R}{P+R}$$
where *P* stands for the precision rate and *R* stands for the recall rate; *P* and *R* are calculated as follows:(24)$$P=\frac{TP}{TP+FP}$$
(25)$$R=\frac{TP}{TP+FN}$$
where TP is the number of entities correctly identified by the model, FP is the number of incorrectly identified entities, TP+FP represents the number of all predicted entities. FN is the number of entities that are false negatives, TP+FN is the number of gold entities.

### 4.3. Parameter Setting

This experiment was developed via PyTorch 1.8.2, using the Python 3.6 language on Win10 operating system with the following hardware environment: Intel (R) Xeon (R) CPU E5-2698 v4 @ 2.20 GHz processor, 16 GB running memory (RAM), GPU for NVIDIA Tesla K80, and 16 G memory. Experimental parameters are as follows: The hidden layer nodes of the FFN layer in the cross-transformer on the Weibo and resume datasets are 384, the dropout parameter is 0.3, the number of multi-head attention layer heads is 8, the dimension of each head is 16, the total number of nodes is 128, the learning rate is set to 0.0018, and the epoch is set to 50. For the MSRA dataset, the number of hidden layer nodes in the FFN layer of the cross-transformer is 480, the dropout parameter is set to 0.3, the number of heads in the multi-head attention layer is 8, the dimension of each head is 20, the total number of nodes is 160, the learning rate is set to 0.0014, and the epoch is set to 100.

## 5. Results

In this section, we compare the different methods applied and discuss the results. To make the experimental results more reasonable, we set up additional working methods for assessing the performance of syntactic information and inner regularity.

### 5.1. The Impacts of Different Improvement Strategies on the F1 Score

The three improved strategies proposed above are, respectively, calculated for the F1 of the model, and the original FLAT model is used as a reference. The results are shown in Table 1, where the baseline is the base model FLAT, KNMN is the first improved strategy added in our model for syntactic information extraction and the endcode, CT represents the cross-transformer module, RP represents the regularity perception module, Kcr-FLAT is the model proposed in this paper.

As can be seen from Table 1, each improved strategy used in this paper improved the performance of the original FLAT model to varying degrees. Among them, adding the KVMN had the most significant improvement of the model, about 6% higher on the Weibo dataset and nearly 2% higher on MSRA and resume datasets because adding the syntactic information improves the accuracy of Chinese NER in different semantic scenarios. By adding the cross-transformer module, the performance was 0.5% higher on MSRA and Weibo datasets, but experienced a 0.25% drop on the resume dataset, as the highly formatted text in this dataset may have limited the performance of the integration of syntactic and lexical information through cross-transformer. The regularity perception module improved the model by 2% on average. The designed Kcr-FLAT Chinese-named entity recognition model was 3% higher than the original Chinese NER model FLAT on the MSRA dataset, 2% higher on the resume dataset, and about 6.5% higher on the Weibo dataset. The addition of syntactic information was fully fused with lexical information by the cross-transformer module, which greatly improved the accuracy of NER in contextually rich datasets, such as Weibo.

### 5.2. Comparison with Other Methods

Taking F1 as the evaluation index, the proposed model was compared with Sui et al. [18], Ding et al. [19], Glyce [23], Xu et al. [20], Ma et al. [21], SLK-NER [22], Nie et al. [26], MECT [24], RICON [27], Zhu et al. [38], Mao et al. [39], and the non-flat-lattice transformer (NFLAT) [40]. The results are shown in Table 2.

Compared with Chinese-named entity recognition models in recent years, the model designed in this paper has obvious advantages in the three different datasets: MSRA, resume, and Weibo. Compared with the RICON model, the designed model does not improve significantly with the MSRA and resume datasets, but with the Weibo dataset, it improves nearly 4% due to the syntactic information encoded by KNMN in the model, which demonstrates the effectiveness of augmenting the input information by syntactic information to improve the recognition accuracy of Chinese-named entities in contextually rich texts. However, the designed model in Weibo data is 2.5% points lower than that of Zhu et al. The boundary-smoothing method of their work helped the trained NER model maintain calibration so that the generated confidence can better represent the accuracy of the prediction entity and have better generalization, thus achieving remarkable results with the Weibo dataset. For the text of variable scenes, regularity may limit the recognition accuracy of named entities in different contexts, but in specific scenarios, the addition of the regularity perception module can improve the recognition accuracy of named entities. For example, the model performs better than MECT, RICON, and Mao et al., on the MSRA and resume datasets.

### 5.3. Case Study

To better understand how our model brings improvement, we conducted a case study on two examples selected from the MSRA and Weibo datasets, respectively. The base model FLAT was used for the comparison and analysis with the designed Kcr-FLAT model. The Chinese-named entity recognition results are shown in the following table.

Where B, M, E represent the beginning, the middle, and the end of an entity, respectively; O shows that the character is not an entity; ORG represents the type of the entity; B-LOC represents the beginning of an entity, which type, the *location*, etc.

Where B-LOC.NAM represents the beginning of a specific location, E-LOC.NOM represents the end of a general location, etc.

As can be seen from Table 3 and Table 4, the designed Chinese-named entity recognition model Kcr-FLAT is improved compared with the original model FLAT. In the example of “*huang jia hai jun fu zhu dui* (Royal Navy Auxiliary Corps)”, the recognition results of FLAT are “*huang jia hai jun* (Royal Navy)” only, and “*fu zhu dui* (Auxiliary Corps)” is ignored since it is not an entity. However, in this case, although “*huang jia hai jun* (Royal Navy)” is identified, it is not the proper entity type according to the context. In regard to syntactic information, both “*huang jia hai jun* (Royal Navys)” and “*fu zhu dui* (Auxiliary Corps)” are the same syntactic components, the weights assigned to their syntactic data are equal. The Kcr-FLAT identifies the whole “*huang jia hai jun fu zhu dui* (Royal Navy Auxiliary Corps)” as an independent entity, which is more accurate.

In the example of “*long zhi meng jiu dian* (longemont hotel)”, the result of FLAT recognition is “*jiu dian* (hotel)”. Due to the limitation of the lexical method, the entity name consisting of single characters that do not form a word is hard to be identified. The proposed model Kcr-FLAT learns the regularity of “XX + *zhi* (of) + XX” due to the regularity perception module paying its highest attention score on thee character “*zhi* (of)”, so that patterns, such as “*long zhi meng* (longemont)”, can be accurately identified, or “*gui guai zhi di* (land of ghost)” in other cases.

## 6. Conclusions

In this paper, we propose a Chinese NER model namely Kcr-FLAT. The proposed method uses a key-value memory network to encode syntactic information, a cross-transformer module to integrate the syntactic information with lexical information, and a regularity perception module to explore the internal regularity of an entity. Experimental results on three datasets demonstrate that the proposed model can effectively solve the segmentation error and improve the performance of Chinese NER. In future research, we will search for a more efficient way to extract syntactic information without other NLP tools, and consider how to integrate external lexical information and internal regularity in a better way to improve the performance of Chinese NER and extend it to other NLP tasks.

## Figures and Tables

**Figure 1 sensors-23-01771-f001:**
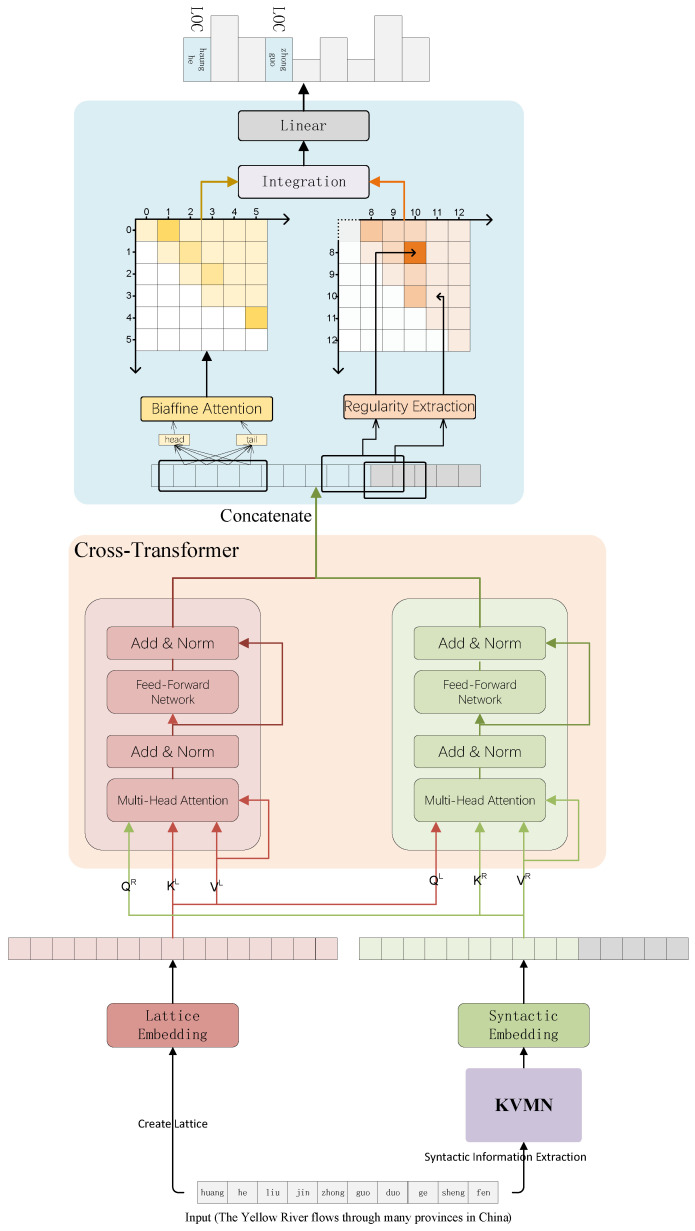
Overall structure of Kcr-FLAT.

**Figure 2 sensors-23-01771-f002:**
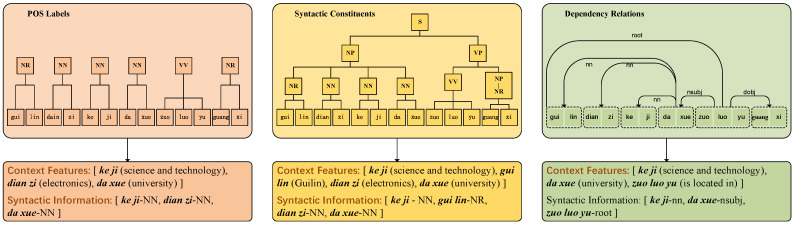
The extracted syntactic information in POS labels, syntactic constituents, and dependency relations for ‘ke’ (science).

**Figure 3 sensors-23-01771-f003:**
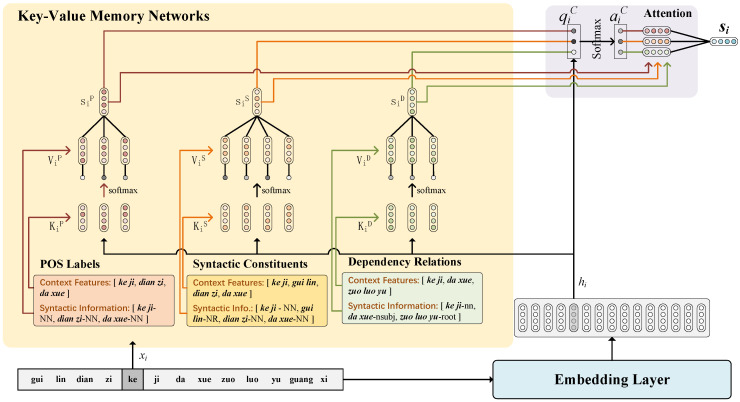
KVMN for encoding syntactic information.

**Figure 4 sensors-23-01771-f004:**
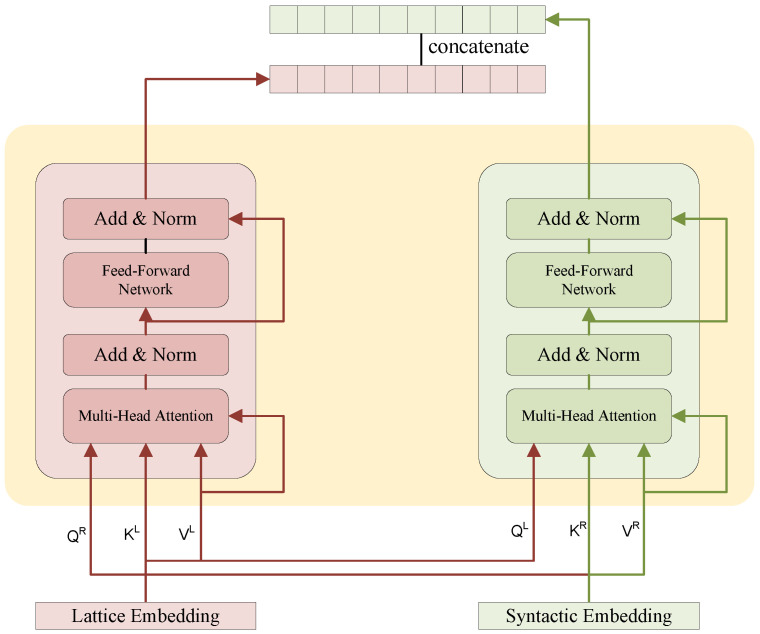
The cross-transformer module.

**Figure 5 sensors-23-01771-f005:**
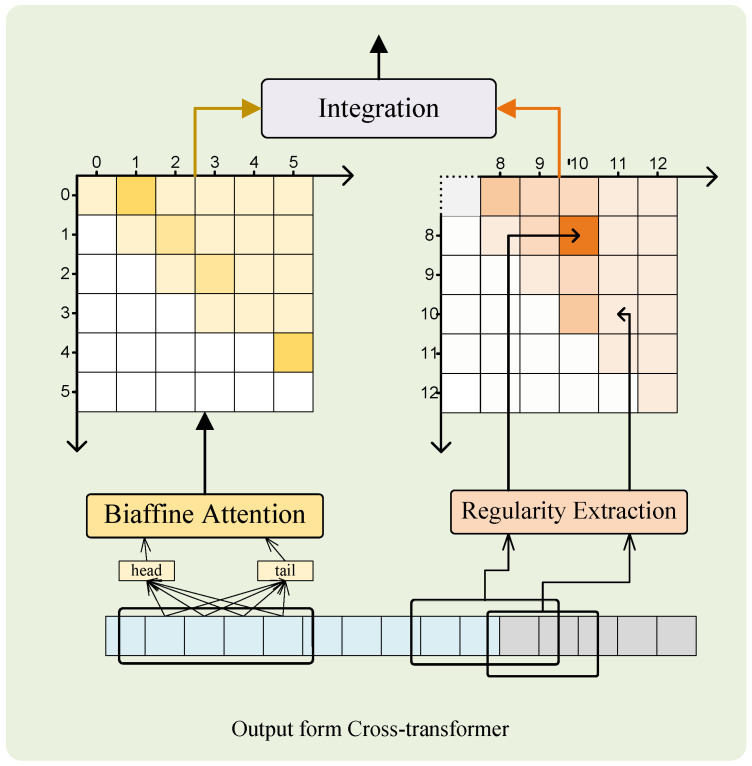
The regularity perception module.

**Table 1 sensors-23-01771-t001:** Ablation experiments.

Model	KVMN	CT	RA	MSRA	Resume	Weibo
Baseline				93.45	94.93	63.42
+KNMN	✓			95.98	96.23	69.27
+KVMN & CT	✓	✓		96.24	95.98	69.78
+RP			✓	96.14	96.02	66.98
Ours (Kcr-FLAT)	✓	✓	✓	96.51	96.81	70.12

**Table 2 sensors-23-01771-t002:** Comparison with other Chinese-named entity recognition models.

Models	MSRA	Resume	Weibo
Sui et al. [18]	-	-	63.09
Ding et al. [19]	-	-	59.50
Glyce [23]	93.26	96.54	67.60
Xu et al. [20]	-	-	68.93
Ma et al. [21]	-	95.59	61.24
SLK-NER [22]	-	95.80	64.00
Nie et al. [26]	81.18	96.62	69.78
MECT [24]	96.24	95.98	70.43
RICON [27]	96.14	96.02	66.89
Zhu et al. [38]	96.26	96.66	**72.66**
Mao et al. [39]	-	96.33	70.85
NFLAT [40]	94.55	95.58	61.94
Ours (Kcr-FLAT)	**96.51**	**96.81**	70.12

**Table 3 sensors-23-01771-t003:** Comparison of Chinese-named entity recognition results on the MSRA dataset.

MSRA	*Huang Jia Hai Jun Fu Zhu Dui* (Royal Navy Auxiliary Corps)						
Entity	huang	jia	hai	jun	fu	zhu	dui
gold label	B-ORG	M-ORG	M-ORG	M-ORG	M-ORG	M-ORG	E-ORG
Baseline (FLAT)	B-ORG	M-ORG	M-ORG	E-ORG	O	O	O
Ours (Kcr-FLAT)	B-ORG	M-ORG	M-ORG	M-ORG	M-ORG	M-ORG	E-ORG

**Table 4 sensors-23-01771-t004:** Comparison of Chinese-named entity recognition results on the Weibo dataset.

Weibo	*Long Zhi Meng Jiu Dian* (Longemont Hotel)				
Entity	long	zhi	meng	jiu	dian
gold label	B-LOC.NAM	M-LOC.NAM	M-LOC.NAM	M-LOC.NAM	E-LOC.NAM
Baseline(FLAT)	O	O	O	B-LOC.NOM	E-LOC.NOM
Ours(Kcr-FLAT)	B-LOC.NAM	M-LOC.NAM	M-LOC.NAM	M-LOC.NAM	E-LOC.NAM

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
