# Peer review of "Kcr-FLAT: A Chinese-Named Entity Recognition Model with Enhanced Semantic Information"

_sensors, 2023, doi:10.3390/s23041771_

Round 1

Reviewer 1 Report

cvThe authors propose a Named Entity Recognition (NER) model for
Chinese, a language highly sensitive to sentential syntactic because
its complex structure in terms of both lexical and syntactic
treatment. In this context, they try to improve the performance of the
task by including enhanced semantic information.

TYPO: using inner ... module --> using a inner ... module

INTRODUCTION: Authors sometimes use acronyms (e.g., NER) and sometimes
do not. Other acronyms such as NLP (for Natural Language Processing)
should be introduced. Please, introduce all the acronyms (as for
example, FLAT, CRF, ...) before use.

The quality of the English is not good (in the whole text), nor is the
writing itself ... for example, the expressions "named entity
recognition" and "Chinese" are continuously repeated. Both aspects
need to be reviewed in depth.

TYPOs: popularity of ... terminal --> popularity of ... terminals

      of Chinese text --> of Chinese texts

      , and text generation --> and text generation

      natural language processing, it --> natural language processing, and it

RELATED WORKS: The authors do not relate the shortcomings and
advantages of previous work to their own proposal, so as to justify
their own proposal. In this context, the Section is perfectly useless
in the contextualization of the work.

TYPOs: in The Sixth --> in the sixth

PROPOSED METHOD: Please, introduce the acronyms (for example, KMMN
... or KVMN ?)  and review the wording of the first five lines. An
intuitive justification of the Eqs. (1-6) in Subsection 3.1 would
be welcome.

Please, introduce and justify intuitively Eqs. (9-11) in Subsection
3.2, and (12-14, 15-19) in Subsection 3.3. It is impossible to
determine the reasons that have inspired the authors.

EXPERIMENTAL SETTINGS: Please, write an introductory paragraph (before
4.1) to explain the testing strategy. Please, include the references
for Pytorch, Python and Win10. Introduce the acronym FFN and justify
the settings.

TYPOs: international Chinese language processing bake off -->
       International Chinese Language Processing Bake-Off

       is proposed by Zhang et al. --> is proposed by Zhang et al. ???

       by Peng et al., --> by Peng et al. ???,

       microblogging --> micro-blogging

       msra --> MSRA

RESULTS: Please, write an introductory paragraph (before
4.1) to explain the structure of this Section.

"Ablation Study ?" ... sorry but I cannot understand the sense of such
title in this context (5.1).

Please, introduce the references in the first sentence of
5.2. Impossible to know which tools the authors are referring to.

In relation to Subsection 5.3, the choice of the case study is
random. The question arises whether this is the best choice strategy
as opposed to one justified on the basis of the linguistic
characteristics of the text. It does not seem, in any case, that the
authors can justify their choice on the basis of the
representativeness of the set of documents finally selected. In this
In this sense, it is worth asking whether the subsequent discussion is
of real use in estimating the performance of the proposal.

CONCLUSION: The choice of the case study (Subsection 5.3)
compromises the authors' conclusions.

Reviewer 2 Report

1. The contribution of the work is not clear. Since the authors used the existing model, therefore, technical contribution in missing.
2. The authors need to highlight what are the gaps left by previous approaches and how proposed framework filled those gaps
3. Please include the significance of this work.

4. Many method improvements of the authors are the integration of a large number of existing methods. Are they simple system integration or have new features? How can the authors prove the effectiveness of the integration methods?
5. The accuracy of the proposed method is high, which is commendable. As everyone knows, increasing the complexity of the network can improve its accuracy. However, the lightweight network and reducing calculation time are also important parts and hot topics in the field of deep learning. What are the improvements of the proposed method in lightweight compared with classical network models? How do you balance the relationship between model accuracy and calculation speed?

6. Author is advised to incorporate the recent research works published in this area.

7. Author is advised to incorporate the recent research work published in this area. Compare and contrast your work with the latest algorithms and principles developed in the past few years.

8. The comparison algorithm is too old. Compare and contrast your work with the latest algorithms and principles developed in the 2022.

Round 2

Reviewer 1 Report

The authors have implemented the suggestions made. The paper has gained a notable improvement in clarity.

Reviewer 2 Report

The manuscript can be accepted.